# The Effects of DNA Methylation on Cytoplasmic Male Sterility in Sugar Beet

**DOI:** 10.3390/ijms25021118

**Published:** 2024-01-17

**Authors:** Jiamin Weng, Hui Wang, Dayou Cheng, Tianjiao Liu, Deyong Zeng, Cuihong Dai, Chengfei Luo

**Affiliations:** 1School of Chemistry and Chemical Engineering, Harbin Institute of Technology, Harbin 150001, China; wengjiamin22@163.com (J.W.); wang_hui@hit.edu.cn (H.W.); sfangelos@163.com (T.L.); cfluo7375@hit.edu.cn (C.L.); 2School of Medicine and Health, Harbin Institute of Technology, Harbin 150001, China; 18b925086@stu.hit.edu.cn (D.Z.); dch@hit.edu.cn (C.D.)

**Keywords:** sugar beet, cytoplasmic male sterility, DNA methylation, whole-genome bisulfite sequencing, methylated gene

## Abstract

DNA methylation is widely found in higher plants and can control gene expression by regulation without changing the DNA sequence. In this study, the whole-genome methylation map of sugar beet was constructed by WGBS (whole-genome bisulfite sequencing) technology, and the results of WGBS were verified by bisulfite transformation, indicating that the results of WGBS technology were reliable. In addition, 12 differential methylation genes (DMGs) were identified, which were related to carbohydrate and energy metabolism, pollen wall development, and endogenous hormone regulation. Quantitative real-time PCR (qRT-PCR) showed that 75% of DMG expression levels showed negative feedback with methylation level, indicating that DNA methylation can affect gene expression to a certain extent. In addition, we found hypermethylation inhibited gene expression, which laid a foundation for further study on the molecular mechanism of DNA methylation at the epigenetic level in sugar beet male sterility.

## 1. Introduction

*Beta vulgaris* L. is an important crop for sugar production, but its yield is currently low [1]. However, the discovery of male sterility in sugar beet by Owen has opened up new possibilities for increasing yields. Two types of male sterility-inducing methods have been identified: genetic male sterility (GMS) and cytoplasmic male sterility (CMS) [2,3]. CMS is a phenomenon inherited maternally and is controlled by the cytoplasm, independent of the nucleus [2]. It is well known that hybrid offspring often display superior traits compared to their parents, including higher yield and increased adaptability [3]. As a result, the production of CMS hybrid seeds has been effectively employed in various crops [4].

CMS in sugar beet is characterized by the presence of abnormal male gametes, while the pistil develops and can accept foreign pollen for fertilization and fruit production [5]. At the cellular level, CMS is caused by dysfunctional interactions between genes in the mitochondria and the nucleus. Previous studies have indicated that CMS is associated with mitochondrial dysfunction, tapetal abnormalities, ATPase malfunction, and altered levels of endogenous hormones. In sugar beet CMS, the pistils and ovules exhibit normal development, but the anthers shrivel and exhibit low vigor [6,7,8]. During the tetrad period, callose enzyme secretion promotes the release of microspores, followed by untimely degradation of the tapetal layer itself. As a result, vacuoles proliferate and form a polynucleated protoplasmic layer, which hinders the timely delivery of water and nutrients. Stagnation of microspore development results in anther abortion [9].

Studies have demonstrated that the ATPase dynamics in anther walls and somatic tissues differ between the CMS line, maintainer line, and restorer line. ATPase is responsible for the hydrolysis of ATP. It provides energy for various cellular processes such as transportation, synthesis, and decomposition of substances. The differences in the distribution of ATPase in anther walls and somatic tissues have been found to impact nutrient transport in the anther. It has a negative influence on pollen in CMS lines [10].

CMS in sugar beet is commonly associated with specific open reading frames (ORFs) found in mitochondrial genes [11]. These ORFs are known to play a significant role in the development of CMS. The occurrence of CMS is facilitated by rearrangements in mitochondrial DNA (mtDNA) and the presence of chimeric genes [12]. Min et al. have identified four ORFs associated with CMS in cotton mtDNA [13]. In some cases, the expression of these new ORFs in mitochondria leads to male sterility or semi-sterility, while in other cases, no such effect is observed. For example, the mitochondrial-targeted expression of *ORF220* in *Brassica juncea* has been found to induce male sterility [12]. Yamamoto et al. conducted a Western blot analysis of the CMS line and maintainer line of sugar beet and discovered an *ATP6* (*preSatp6*)-encoded 35KDa nucleotide fragment, which may be a polypeptide related to male sterility. Additionally, the 12 KDa polypeptide encoded by *ORF129* has been implicated in affecting male sterility. It is speculated that *ORF129* may promote the expression of the *COX2* in plants, which impairs mitochondrial function and fails to provide the necessary energy for flower bud development [14].

Various factors can contribute to CMS, and the development of anthers involves intricate metabolic pathways. One crucial epigenetic modification is DNA methylation. It occurs mainly at cytosines in the CG, CHG, and CHH contexts. Numerous studies have reported associations between DNA methylation and CMS [15,16,17]. For instance, Li et al. employed WGBS to analyze the DNA methylation levels in young panicles of cultivated and wild rice. The regions with methylation modifications exhibited lower expression levels compared to unmethylated regions [18]. Additionally, research has demonstrated higher levels of DNA methylation polymorphisms in the F1 generation than in the parents. It indicates a relationship between DNA methylation and different cytoplasmic types [19,20]. In cotton, DNA methylation genes are concentrated in vital metabolic pathways, including the starch and sucrose metabolism pathways and galactose metabolism [21]. In the soybean CMS line, pollen abortion is associated with changes in DNA methylation genes involved in carbohydrate and energy metabolism, transcriptional regulation, male gametophyte growth, and mitochondrial proteins [22]. However, the relationship between the expression patterns of DNA methylation and CMS in sugar beet remains unclear. Therefore, this study aims to identify differentially methylated genes between CMS lines and maintainer lines in sugar beet through WGBS technology. Furthermore, functional analysis will be conducted to explore the molecular mechanism underlying CMS in sugar beet.

## 2. Results

### 2.1. DNA Methylation Pattern of Sugar Beet

Genomic DNA libraries were constructed using flower buds collected from the sugar beet CMS line DY5CMS and its maintainer line DY5O, aiming to investigate the genome-wide DNA methylation pattern of sugar beet. Each beet flower bud sample generated an average of 26.699 Gb of clean data after filtering. The BSMAP(ver2.90) software was employed to compare the high-quality read segments obtained with *Beta vulgaris* L., and the statistical results of the comparison are presented in Table 1. The analysis revealed that over 99% of the DNA in sugar beet buds was methylated, indicating a high recognition rate of the WGBS method. The reads from each beet bud DNA library were further subjected to de-duplication processing as required. Quality control was performed to ensure the sequencing data met the required standards. The clean base counts for DY5O and DY5CMS were 248,189,440 (86.16%) and 301,409,704 (86.94%). These results demonstrate the high reliability and accuracy of the method used in this study.

Table 2 displays the methylation levels of DY5CMS and DY5O, which were found to be 27.32% and 27.18%. These levels were consistent with the reported DNA methylation range of 6% to 30% in the whole plant genome. Specifically, the DNA methylation levels were observed to be 80.43% and 79.96% in the CG context, 58.72% and 58.69% in the CHG context, and 12.42% and 12.28% in the CHH context for DY5CMS and DY5O. In the three contexts, the methylation level of CMS line DY5CMS was close to that of maintainer line DY5O. The highest methylation level was observed in the CG context.

To explore the relationship between sequence characteristics and DNA methylation bias, the methylation percentage of nine bases surrounding the methylation sites was calculated. Figure 1 illustrates that cytosine methylation primarily occurs in the CG context, typically within the TCGA context. In the CHG context, methylation sites are frequently found in the CTG context, followed by the CAG context, and less commonly in the CCG context. In the CHH context, methylation predominantly occurs in the CAA context, with the CTA context being the next preference, and finally, in the CCC context. These findings suggest that the preference for cytosine methylation context remains consistent in both the CMS line and maintainer line of the sugar beet genome.

### 2.2. DNA Methylation Level Analysis of DY5CMS and DY5O

In plants, DNA methylation occurs not only in the CG context but also in the CHG and CHH contexts. In our study, we found that the genome-wide DNA methylation levels of the sugar beet CMS line and maintainer line were similar, with no significant difference in methylation levels. However, the CMS line DY5CMS had relatively higher methylation levels. Figure 2 displays the distribution of methylated cytosines in the sugar beet CMS lines. Among them, 26.29% of methylated cytosines are found in the CG context, 22.76% in the CHG context, and 50.94% in the CHH context. Similarly, in the sugar beet maintainer lines, the distribution of methylated cytosines includes 27.47% in the mCG context, 23.66% in the mCHG context, and 48.86% in the mCHH context. Notably, the distribution patterns of methylated cytosines in the mCG, mCHG, and mCHH contexts were consistent between the sugar beet CMS lines and maintainer lines.

Different types of C base (mCG, mCHG, and mCHH) DNA methylation levels were different among different species and even under different cell types of the same species. The methylated cytosine content of the two sugar beet samples was different, but the number of methylated cytosine sites was similar between the two cohorts. In Figure 3A, in DY5CMS, 8,804,806 mC were found in the coding sequence (CDS) region, while 13,618,804 and 13,317,846 mC were located in the upstream and downstream 2000 bp regions of the transcript. Additionally, 46,366,216 mC were present in the mRNA region, and 19,546,788 mC belonged to the repeat region. Furthermore, 6,297,583 mC were situated on CpG islands. In Figure 3B, 8,788,857 mC were found in the CDs region, while 13,616,831 and 13,304,356 mC were situated in the 2000 bp regions upstream and downstream of the transcript. Moreover, 46,348,605 mC were present in the mRNA region, and 19,723,251 mC belonged to the repeat region. Additionally, 6,308,297 mC were situated on CpG islands.

As shown in Figure 3C,D, among the six different gene functional elements, the average methylation levels of repeat and CpG island regions were the highest. This suggests that DNA methylation primarily occurred in these repeats and CpG island regions, which may play a role in epigenetic regulation. Furthermore, it can be concluded that the majority of methylated cytosine is located in the mRNA region. Comparing different gene elements, the cytosine quantity of mC in DY5CMS is generally higher than that in DY5O. Additionally, the average methylation level in the mRNA region was the lowest, which suggests a possible connection with male sterility in the CMS line. It is speculated that the transcriptional expression of genes associated with CMS may be influenced by reducing the degree of gene methylation.

Figure 4 illustrates the distribution of methylation levels for each type of methylated cytosine (CG, CHG, and CHH) in sugar beet, providing insights into the DNA methylation characteristics. In both CG and CHG contexts, higher levels of methylation were observed in the upstream, first intron, internal intron, and downstream regions. The degree of methylation is more pronounced in the CG context compared to the CHG and CHH contexts, and the CHH context has the lowest methylation level in each gene region.

### 2.3. Verification of WGBS Sequencing Data by Bisulfite Conversion Method

Based on the WGBS analysis of DY5CMS and DY5O, three differential methylation regions (DMRs) were randomly chosen for further investigation. Among these DMRs, two regions exhibited hypermethylation, while one displayed hypomethylation. The purpose of analyzing these regions was to validate the accuracy of the WGBS data. In Table 3, the methylation levels of methylcytosine (mC) in the three selected regions are presented, comparing the CMS line and the maintainer line across different methylation contexts.

As depicted in Figure 5, our analysis of the *BVRB_7g*157750 revealed that the methylation level of the sugar beet maintainer line was higher than that of the CMS line. This was observed through sulfite treatment and WGBS. Similarly, for the DMGs *BVRB_5g*121390 and *BVRB_1g*006690, sulfite treatment and WGBS indicated that the methylation level of the sugar beet CMS line was higher than that of the maintainer line. The results obtained from bisulfite treatment in three different methylation regions were consistent with the WGBS data. It showed a 100% coincidence rate between the sulfite treatment results and WGBS data. This high level of agreement suggests that the data obtained through WGBS is reliable.

### 2.4. Differential Methylation Genes (DMGs) Identification of DY5CMS and DY5O

Based on the results of WGBS, we conducted a screening of genes with differential methylation (DMGs). We selected DMGs with differentially methylated regions (DMR) that have a length of at least 150 base pairs and with a *p*-value less than 0.001 for further analysis. In the comparison between DY5O and DY5CMS, we classified the up-regulated DMGs in DY5CMS as hyper-DMGs and the down-regulated DMGs as hypo-DMGs. When the DMGs overlap in different gene functional regions, they show both hypermethylation and hypomethylation, and it is called instability-DMG.

In Figure 6, we categorized the differentially methylated regions (DMRs) into two groups: DMR-related genes and DMR-related promoters. From the WGBS results of DY5CMS and DY5O, we identified a total of 2117 DMRs. Among these, 936 genes showed high methylation levels, while 591 genes displayed low methylation levels. When it comes to the promoters, we found 252 DMRs that were highly methylated and 192 DMRs that were hypomethylated. Additionally, we observed a total of 146 DMRs in both promoters and related genes. Among these, 50 DMRs exhibited hypermethylation, 60 DMRs showed hypomethylation, and 36 instability-DMRs. These results indicate that most of the related genes and promoters of DY5CMS have higher methylation levels than DY5O.

### 2.5. GO Annotation and KEGG Enrichment Analysis of Sugar Beet DMGs

To better understand the molecular functions of the differentially methylated genes (DMGs) identified in CMS lines and maintainers of sugar beet, a GO enrichment analysis was performed. This analysis revealed that DMGs are involved in 34 functional categories, including 18 biological processes (BPs), three cell compositions (CCs), and 13 molecular functions (MFs). In terms of BPs, the DMGs are primarily associated with cellular processes and metabolic processes. They also exhibit catalytic activity and binding in the molecular functions category. In the CC category, DMGs were mainly enriched in biofilm and cytoplasmic parts. Furthermore, the MF category revealed that the DMGs are predominantly involved in catalytic activity and transcription processes. KEGG pathway enrichment analysis (*p* < 0.05, Q < 0.05) identified 20 biological metabolic pathways in which the DMGs are implicated. These pathways include metabolic processes such as the metabolic pathway (KO01100), glycolysis pathway (KO00010), metabolism of starch and sucrose (KO00500), plant hormone signal transduction (KO04075), biosynthesis of secondary metabolites (KO01110), and biosynthesis of amino acids (KO01230) (Figure 7).

Combined with GO functional annotation and KEGG pathway enrichment analysis, 12 genes that may be related to sugar beet CMS were selected by referring to relevant reports in the literature (Table 4). These DMGs were flowering regulation gene *BVRB_6g*133580, flower organ development regulation gene *BVRB_3g*061680, transcription-related genes *BVRB_2g*032900 and *BVRB_6g*135030, plant cell wall synthesis-related gene *BVRB_6g*137940, endogenous hormone regulation-related gene *BVRB_003480*, ATPase activation-related gene *BVRB_4g*073090, and mitochondrial structure-related genes *BVRB_2g*036570 and *BVRB_*004020.

### 2.6. Analysis of the Relationship between DMGs’ Methylation and Expression Level’

Twelve DMGs related to sugar beet fertility were screened between DY5CMS and DY5O using WGBS technology combined with GO annotation and pathway analysis. Transcriptional expression levels of 12 DMGs were analyzed by quantitative real-time PCR (qRT-PCR) to further study the relationship between gene methylation level and expression level. The expression level of 25% of genes was positive feedback to the degree of methylation, while 75% of genes were negative feedback to the degree of methylation. DNA methylation modification could affect the transcription expression of some genes to a certain extent.

As shown in Appendix A and Appendix A, ABC transcription factor protein gene *BVRB_9g*208010, WRKY transcription factor protein gene *BVRB_4g*075270, cysteamine protease gene *BVRB_6g*137940, ACO enzyme gene *BVRB_3g*049380, and other genes had higher methylation degree in sugar beet CMS lines. The transcriptional expression level was lower than that of the maintainers. UDP-glucuronic acid decarboxylase gene *BVRB_7g*164820, WRKY transcription factor protein gene *BVRB_6g*133580, and TDR transcription factor protein gene *BVRB_7g*172330 had low methylation levels and increased transcription expression levels in sugar beet CMS lines. Therefore, it is speculated that the methylation level of plants may affect the transcriptional expression level, and genes in the hypermethylated state may inhibit gene expression, leading to down-regulation of the expression level. When the methylation level of the gene was decreased, its expression level was up-regulated. It was found that NADH dehydrogenase gene *BVRB_2g*047050 was hypermethylated in sugar beet CMS lines, but its expression level was up-regulated, and DNA methylation modification did not play a role, which needed further study.

## 3. Discussion

DNA methylation is widely found in higher plants. It affects DNA repair and gene transcription. At present, the genome-wide DNA methylation maps of Arabidopsis, soybean, cotton, and other crops have been reported successively, but the map of sugar beet has not been learned [21,23,24]. This paper performed genome-wide methylation analysis between sugar beet CMS line DY5CMS and its maintainer line DY5O through WGBS technology. DNA methylation occurs in CG, CHH, and CHG contexts. The methylation level in the CG context was significantly higher than that in the other contexts. A total of 2117 DMRs were identified between DY5CMS and DY5O at the tetrad period. Furthermore, 12 DMGs were found to be associated with potential function in the CMS line. The connection between 12 DMGs and pollen abortion in sugar beet was discussed.

### 3.1. Related DMGs Involved in Carbohydrate and Energy Metabolism

Anther tapetum cell plays a crucial role. In the cell meiosis period, programmed cell death (PCD) promotes the degradation of tapetum. Abnormal PCD can lead to tetrad anomaly and pollen abortion. *TDR* is a pivotal component of the transcriptional regulatory network. It regulates tapetal development and degradation [25]. A hypomethylation gene, *BVRB_7g172330,* was found to be homologous with *TDR* from rice. The positive expression of this gene may affect tapetal development, leading to anther abortion in DY-5CMS.

Pyruvate dehydrogenase consists of E1, E2, and E3. It controls the entry of carbon into the tricarboxylic acid cycle to produce energy. In *Arabidopsis*, *MAB1* encodes mitochondrial PDH, leading to decreased pyruvate dehydrogenase activity and metabolic abnormalities [26]. In this study, we found a DMG, *BVRB_004020,* was related to pyruvate dehydrogenase E1. The methylation level of the *BVRB_004020* was higher in DY5CMS than in DY5O. *BVRB_004020* expression significantly decreased in DY5CMS, which resulted in a decrease in the energy generated in the TCA cycle. The energy required for microspore maturation was lacking. It resulted in microspore malformation and then had a negative effect on pollen development.

### 3.2. Related DMGs Involved in Pollen Wall Development

The pollen wall of plants is composed of the inner wall and the outer wall. The inner wall mainly contains cellulose, and the outer wall contains sporopollen, fatty acid derivatives, and phenylpropane. The outer layer of the pollen wall is mainly composed of sterols and contains a small number of flavonoids and alkanes. The ABCG subfamily is involved in lipid transport, pollen development, and inflorescence growth. In *Arabidopsis*, *ABCG15* mainly transports lipid substances required by the cuticle of petal cells. It affects the normal elongation of petals. In rice, Qin Peng et al. found that *ABCG15* of the ABC family is critical to anther and pollen development after meiosis. The anther of the *ABCG15* mutant is white and small, without mature pollen. It is an abnormal development of cuticles and pollen outer wall [27]. In this study, the DMG *BVRB_4g*073090 was found to be orthologous to the *ABCG15* in *Arabidopsis*. The methylation level of this gene in DY5CMS was lower than that in DY5O, and its expression level was higher than that in DY5O. It is speculated that the *ABCG15* is overexpressed, and defects occur in the development of the pollen wall, which affects the normal development of anther and leads to male sterility in DY5CMS.

Galacturonic dehydrogenase (GAUT) belongs to the glycosyltransferase family and is involved in cell wall synthesis, pollen tube synthesis, and anther development in plants. *GAUT1* is the first functional pectin HG-galactosyl transferase (GalAT) discovered in *Arabidopsis*, which is closely related to pectin synthesis [28]. In *Arabidopsis*, the *GAUT1* family encodes 14 GAUT and 10 GATL proteins. It had 56–84% and 4–53% homology with *GAUT1*. Brown et al. found that the contents of xylan and galuronic acid in the anthers of *GAUT12* mutants were reduced, which resulted in pollen failure to develop normally. In *Arabidopsis*, *GAUT13* and *GAUT14* double mutants, pollen tube growth was abnormal, with the pollen tube swelling was normal elongation, and there was no fibrin in the outer cell wall. Defects in pollen tubes in double mutants lead to fertility loss in male gametes [29]. In this study, *BVRB_6g*137940 was identified as a *GAUT1*. It presented a higher methylation. The expression of that DMG was significantly decreased, which may have indirectly affected the development of the pollen tube and anther, resulting in male sterility in DY5CMS.

### 3.3. Related DMGs with Transcriptional Regulation

The flowering time of the *WRKY13* mutant is delayed, and the time of plant transition from vegetative growth to reproductive growth is delayed, resulting in pollen development defects in *Arabidopsis* [30]. In contrast to *WRKY13*, *WRKY12* mutant plants exhibit an early flowering phenotype. It was found that *WRKY12* and *WRKY13* interact with *SPL10* to jointly regulate flowering by age pathway. *SPL10* is a transcription factor with dual regulatory functions, which can both activate *WRKY12* expression and inhibit *WRKY13* expression. In this study, we found that *BVRB_6g*133580 was homologous to *WRKY13* in *Arabidopsis*, its methylation modification in DY5CMS was reduced, and its transcription expression level was increased. It is speculated that *WRKY13* overexpression may lead to delayed flowering time, abnormal pollen development, and male sterility.

The MAD-Box protein family mainly regulates plant growth and development and signal transduction. *AGL15* belongs to the *MIKC* protein of MADS-Box family. *AGL15* mutants of *Arabidopsis* show a large number of morphological changes: reduced leaf morphological fertility, delayed flowering, delayed flower organ shedding, and senescence. Studies have found that *AGL15* can affect the expression of miRNA156, which inhibits the transcription of *squamosa promoter binding protein-like 3*(*SPL3*), resulting in delayed flowering in the plants. *AGL15* is mainly expressed in embryos during early development of *Arabidopsis*, and *AGL15* is expressed in the leaf apical meristem and flower base after germination of *Arabidopsis* [31]. *BVRB_3g*061680 was identified to be highly homologous with *Arabidopsis AGL15*, and its methylation modification in DY5CMS was weakened, and its expression level was increased. It was speculated that *AGL15* might affect the flowering regulation mechanism by affecting miRNA expression, thus leading to delayed flowering.

### 3.4. DMGs Associated with Endogenous Hormones

Endogenous hormones in plants are involved in all physiological processes of plant growth and are closely related to the male sterility of plants [32]. Endogenous hormones mainly include auxin, gibberellin, ethylene abscisic acid, and other ethylene, which are involved in the growth and development of plants, such as programmed cell death of fruit maturation and senescence, and are also related to pollen abortion of plants. It was found that ethylene concentration in the anthers of maize microspores before abortion was significantly higher than that in the maintainers, and excessive ethylene release in CMS lines was closely related to male sterility. The rate-limiting enzymes of ethylene synthesis in plants mainly include ACC synthase (ACS) and ACC oxidase (ACO). In this study, the *BVRB_3g*049380 was identified to be homologous to the *ACO* gene of *Arabidopsis*. The methylation modification of this gene was weakened in sugar beet CMS lines, but its expression level was also reduced in CMS lines. The methylation modification may not affect gene expression, which needs further study.

## 4. Materials and Methods

### 4.1. Plant Materials

Plant experimental materials were all from the self-bred beet DY5CMS and DY5O from Harbin Institute of Technology. All materials were beet biennial crops. Beet buds in the tetrad stage were selected and frozen in liquid nitrogen and stored at −80 °C for further determination.

### 4.2. DNA Extraction and WGBS

Extraction of total genomic DNA was carried out using beet bud by Cetyltrimethylammonium Bromide (CTAB). After quality testing, the genomic DNA was broken into fragments with an average size of 250 bp using Bioruptor (Diagenode, Seraing, Belgium). A base group was added to the 3′ end of the DNA fragment to connect the methylation connector. EZ DNA Methylation Gold Kit (ZYMO, Irvine, CA, USA) was used for Bisulfite treatment. Agar-gel electrophoresis was performed to select fragments in the range of 320–420 bp. DNA fragments were extracted using the QIA Quick Gel Extraction Kit (Qiagen, Valencia, CA, USA) and amplified by PCR. The constructed libraries were tested for quality and yield using Agilent 2100 Bioanalyzer (Agilent, Santa Clara, CA, USA) and ABI StepOnePlus Real-Time PCR System (Thermo, Waltham, MA, USA). Qualified libraries were sequenced on the Illumina sequencing platform. The above WGBS was commissioned by BGI (Shenzhen, China).

### 4.3. WGBS Data Validation

After WGBS sequencing results were completed, EZ DNA Methylation Gold Kit (Zymo Research, CA, USA) was used to treat DY5CMS and DY5O flower bud DNA with bisulfite, and 3 methylation regions with different coverage levels were randomly selected and primers were designed using MethPrimer. (http://www.urogene.org/cgi-bin/methprimer/methprimer.cgi/ accessed on 11 October 2022 online design primers (refer to Table 5). PCR reaction system (50 μL):25 μL RTAP enzyme, 1 μL upstream and downstream primers, and 2 μL treated DNA template were used for Gel recovery using the Mini BEST Agarose Gel DNA Extraction Kit(TaKaRa, Osaka, Japan). The purified product was connected to the PMD18-T vector. A total of 10–12 positive clones were screened and sent to Ruibo Biotechnology Co.LTD for sequencing, and the sequencing results were analyzed by the online analysis software Kismeth (http://katahdin.mssm.edu/kismeth/primer_design.pl) for methylation level.

### 4.4. Evaluation of Methylation Level at Site C

According to the sequencing results, the number and proportion of methylated C sites in each sample under different sequence environments were counted. For the identified methylation sites, the methylation level was calculated. The genome sequence is first divided into many 10 kb intervals (bins), and the number of reads with methylated C (mC) and unmethylated C (umC) within each interval is calculated. The methylation level of each interval or site C is
ML (C) = reads (mC)/[reads (mC) + reads (umC)].

### 4.5. Differential Methylation Regions (DMRS) and Differential Methylation Genes (DMGS) Were Identified

Differentially methylated region (DMR) refers to a region of the genome that is significantly different in methylation degree between two samples and is an important marker of genetic change. A window containing at least 5 CG (CHG or CHH) at the same location of the genome of the two samples was searched, and the difference in CG methylation levels of this window in the data of the two samples was compared. The region with a significant difference in methylation (2-fold difference and Fisher’s test *p* ≤ 0.05) in the two samples was identified as DMR. If the continuous region formed by two adjacent DMRs is significantly different in methylation levels in the two samples, the two DMRs will be combined into one continuous DMR; otherwise, they will be two independent DMRs.

DMRs are widely distributed in promoters, exons, and introns in the genome, so there may be a variety of regulatory gene modes. Firstly, the genomic location of the DMRs and the gene structure annotation information of the reference genome were used for structural annotation and number statistics. When the gene region in which the DMRs reside overlaps with a specific functional element on the reference gene region, it is called DMR-related genes or DMGs.

### 4.6. GO and KEGG Enrichment Analysis of Differential Methylated Genes

GO enrichment analysis provides all GO terms that were obviously enriched in DMR related genes, and DMR related genes with special functions can be found. DMR related genes were compared with terms in the GO database, the number of genes in each term was calculated, and hypergeometric tests were applied (http://www.geneontology.org/ accessed on 12 October 2022). The pathway-based analysis provides further insight into the metabolic pathways in which certain genes are involved. KEGG pathway was used to identify pathways that were significantly enriched in DMR-related genes compared with the whole-genome background (http://www.genome.jp/kegg/ accessed on 12 October 2022).

### 4.7. qRT-PCR Analysis

DMGs were randomly selected from sequencing data, and NCBI was used to design specific primers online for fluorescence quantitative PCR to study the relationship between the expression level and the degree of DNA methylation (http://www.ncbi.nlm.nih.gov/ accessed on 12 October 2022).

## 5. Conclusions

In this investigation, we employed whole-genome bisulfite sequencing (WGBS) to explore the genome-wide DNA methylation patterns of DY5CMS and DY5O. Our findings reveal that DNA methylation predominantly occurs in both CG and non-CG contexts within the beet genomes. Notably, a comprehensive analysis identified a total of 2117 differential methylation regions (DMRs) between the two cohorts. Utilizing GO annotation and KEGG analysis, we pinpointed twelve crucial differentially methylated genes (DMGs) potentially associated with male sterility. These identified DMGs are implicated in processes such as carbohydrate and energy metabolism, pollen wall development, and endogenous hormone regulation. Through quantitative real-time PCR (qRT-PCR), we observed a negative correlation between the methylation levels and the expression of nine DMGs. It indicated that DNA methylation modification can affect gene expression to a certain extent, and hypermethylation modification can inhibit gene expression. These insights contribute to the foundational understanding of DNA methylation alterations and their potential roles in sugar beet cytoplasmic male sterility (CMS), helping further investigations into the intricate mechanisms underlying these modifications.

## Figures and Tables

**Figure 1 ijms-25-01118-f001:**
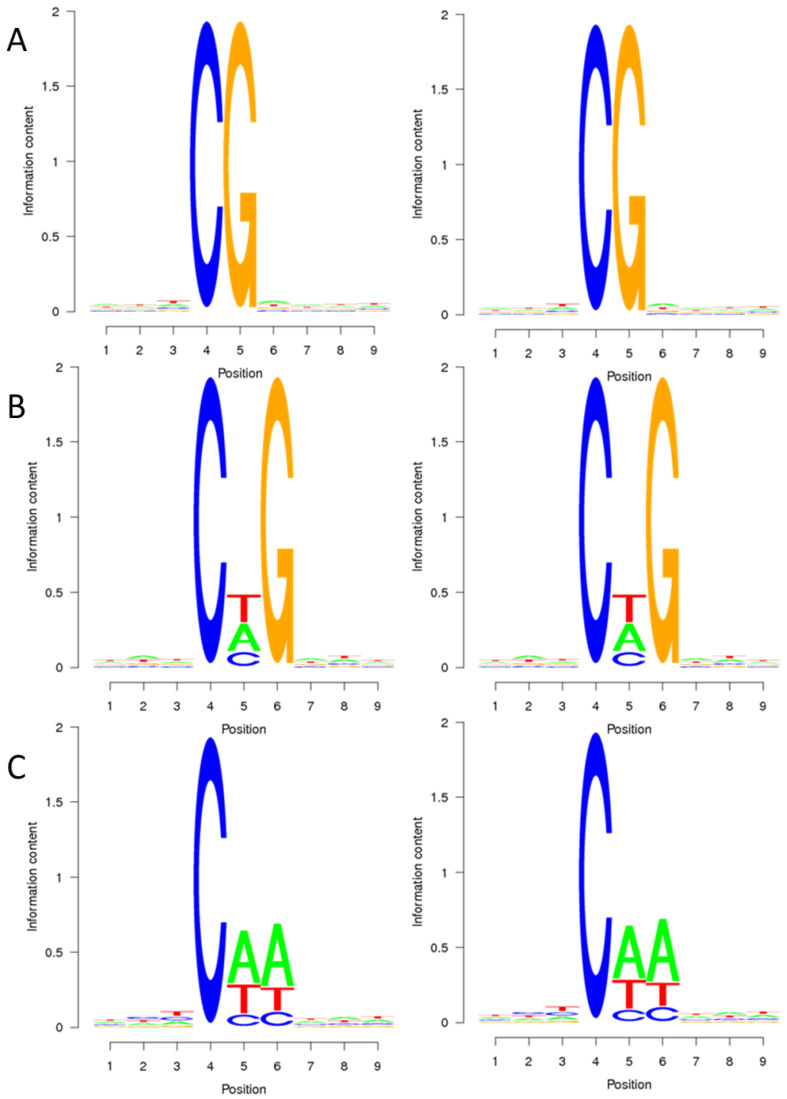
Sequence preference of sugar beet anther methylated cytosine. (**A**–**C**) are the sequence preferences of DY5CMS and DY5O in CG, CHG, and CHH contexts. The horizontal axis represents the base position, with the fourth position being the C base for analysis. The vertical axis is the entropy value (0 is the minimum value, indicating that the four bases are in uniform proportion, 25%, and 2 is the maximum value).

**Figure 2 ijms-25-01118-f002:**
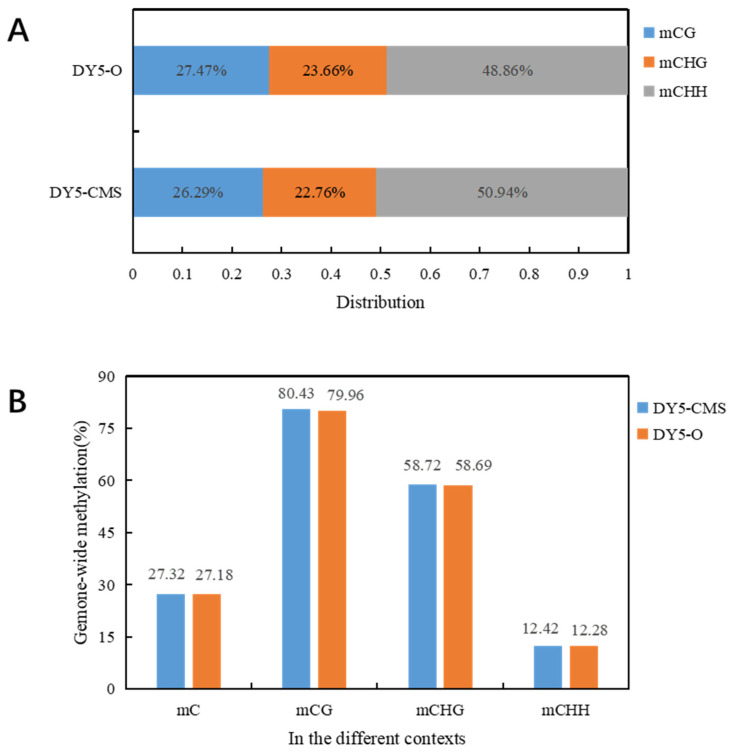
Distribution of methylated cytosine in sugar beet. (**A**) Distribution map of mC in DY5CMS and DY5O under three contexts; (**B**) genome-wide mC level and proportion of mC level in CG, CHG, and CHH context.

**Figure 3 ijms-25-01118-f003:**
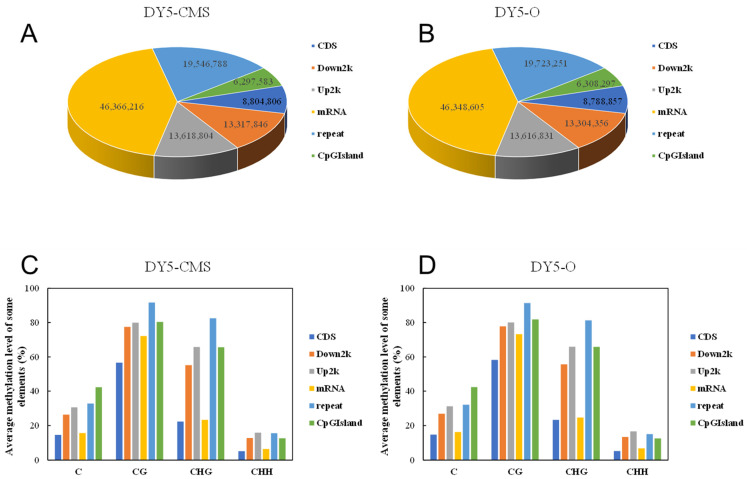
Methylated cytosine distribution of different transcription elements in sugar beet. (**A**) The amount of methylated cytosine in different transcription element regions in DY5CMS; (**B**) the amount of methylated cytosine in different transcription element regions in DY5O; (**C**) the methylation levels in different transcription element regions in DY5CMS; (**D**) the methylation levels in different transcription element regions in DY5O.

**Figure 4 ijms-25-01118-f004:**
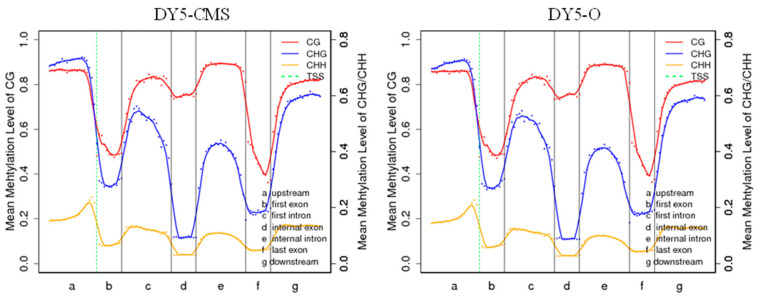
Methylation trend of different transcription element regions in sugar beet. Note: On the horizontal axis, the entire gene is divided into seven different transcription element regions; on the vertical axis, there is the methylation level; and on the dotted green line, there is the TSS (transcription start site) position.

**Figure 5 ijms-25-01118-f005:**
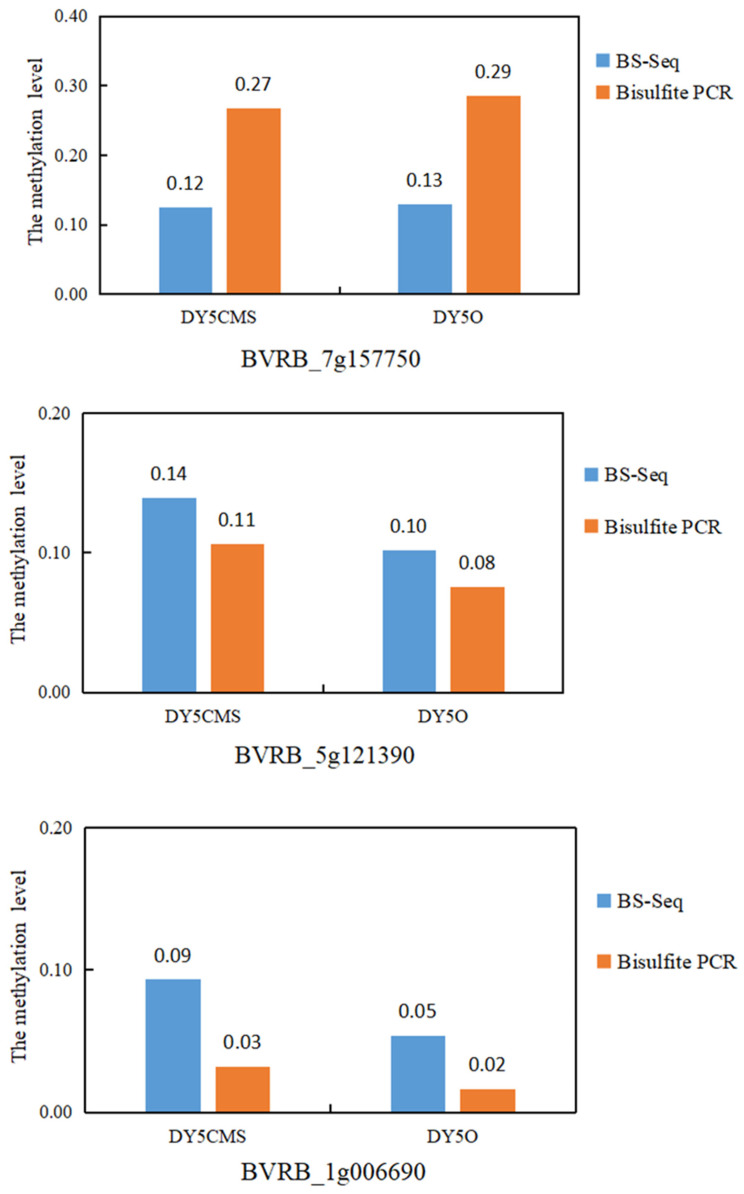
Pattern analysis of three methylation regions in sugar beet.

**Figure 6 ijms-25-01118-f006:**
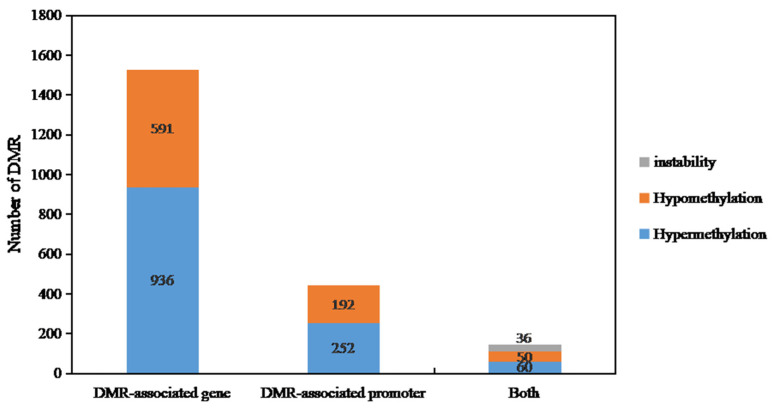
Distribution of differential methylation genes in sugar beet CMS lines and maintainers.

**Figure 7 ijms-25-01118-f007:**
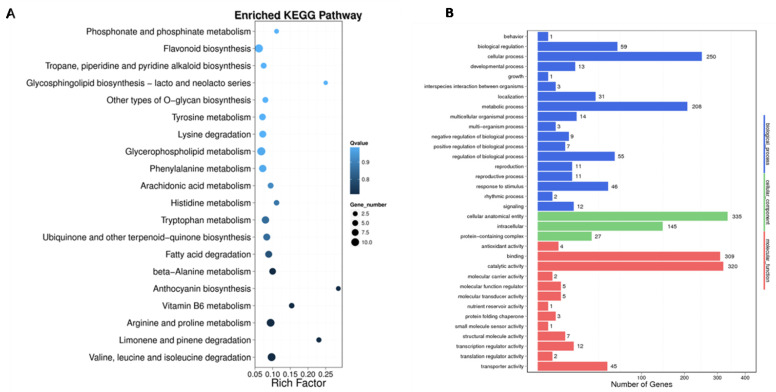
KEGG analysis diagram and GO annotation. (**A**) KEGG analysis diagram of DMGs between DY5CMS and DY5O; (**B**) GO annotation of DMGs between DY5CMS and DY5O.

**Table 1 ijms-25-01118-t001:** Sequence reads were generated by WGBS in DY5O and DY5CMS.

Sample	Clean Bases	Mapped Reads	Mapping Rate (%)	Average Depth (X)	Bisulfite Conversion Rate (%)
DY5O	248,189,440	213,850,619	86.16	29.58	99.14
DY5CMS	301,409,704	262,053,058	86.94	36.00	99.33

**Table 2 ijms-25-01118-t002:** Statistical results of DNA methylation level in sugar beet anther.

Background	Sample	Number of Methylated Cytosine	Mean Methylation Level (%)
C	DY5CMS	53,612,497	27.32
	DY5O	51,332,141	27.18
CG	DY5CMS	14,155,733	80.43
	DY5O	14,101,006	79.96
CHG	DY5CMS	12,223,221	58.72
	DY5O	12,149,595	58.69
CHH	DY5CMS	27,233,543	12.42
	DY5O	25,081,540	12.28

**Table 3 ijms-25-01118-t003:** Sugar beet DNA methylation levels.

Target Area	Sample	DNA Total Methylation Level (%)	mCG (%)	mCHG (%)	mCHH (%)
Chr7: 1,633,239–1,633,426	CMS	12	57.94	4.5	1
	O	13	62.05	36.46	20.45
Chr3: 51,195,686–51,195,932	CMS	14	85.08	37.21	1.98
	O	10	69.42	0	2.02
Chr1: 7,437,126–7,437,376	CMS	9	16.23	66.57	6.7
	O	5	3.05	6.27	6.23

**Table 4 ijms-25-01118-t004:** Identified 12 genes potentially related to sugar beet CMS.

Gene Name	Location	Methylation Status	Gene Annotation
*BVRB_3g*061680	gene	Hypo	MADS-box transcription factor family protein
*BVRB_6g*133580	gene, prompter	Hypo	WRKY transcription factor family protein
*BVRB_2g*032900	promoter	Hyper	AP2/ERF and B3 domain-containing transcription factor family protein
*BVRB_2g*047050	gene	Hyper	NADH dehydrogenase
*BVRB_6g*137940	gene	Hyper	galacturonosyltransferase
*BVRB_*003480	gene	Hypo	UDP-Glycosyltransferase superfamily protein
*BVRB_*004020	gene, prompter	Hyper	pyruvate dehydrogenase E1 component subunit beta-3
*BVRB_9g*208010	gene	Hyper	ABC transporter A family protein
*BVRB_4g*073090	gene	Hypo	ABC transporter G family protein
*BVRB_2g*036570	gene	Hypo	PPR superfamily protein
*BVRB_3g*049380	gene	Hypo	ACO family protein
*BVRB_7g*172330	promoter	Hypo	TDR transcription factor family protein

**Table 5 ijms-25-01118-t005:** Primers required for bisulfite verification.

Primer	Primer Sequence	Location	Length (bp)
Primer1	F:TATTATTTTAAGGGATGGATGTTTTG	Chr7	228
	R:ACTCCAAATTCACAACTATAACCTTCT	1,633,303–1,633,462	
Primer2	F:TGGGTTGATTTAGGTTTATTTGAAT	Chr5	249
	R:ACAAAAAATACATACCCCCTCAAT	51,195,686–51,195,932	
Primer3	F:TGGGGTTAATAATTGTTAGTTTTAT	Chr1	249
	R:TCACCACCCCATCTAAAATATATAC	7,437,126–7,437,376	

## Data Availability

Data is contained within the article and Appendix A.

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
