# Peer review of "The Effects of DNA Methylation on Cytoplasmic Male Sterility in Sugar Beet"

_ijms, 2024, doi:10.3390/ijms25021118_

Round 1
Reviewer 1 Report
Comments and Suggestions for Authors
In the research article entitled “Effects of DNA methylation on cytoplasmic male sterility in Sugar beet” authors constructed whole genome methylation map of sugar beet using whole genome methylation sequencing technology. 12 differential methylation genes (DMGs) were identified, which were related to anther cell wall development, flowering stage, transcription factors, which laid a foundation for further study on the molecular mechanism of DNA methylation at the epigenetic level in sugar beet male sterility etc. The study was conducted systematically in a planned way but I have few minor corrections in the MS before its publication in IJMS.
Line 16: Write full form of WGBS first time
Line 27: write the scientific name of Sugar beet
Line 33-34: delete; “CMS refers to the phenomenon that the male reproductive organs cannot produce normal functioning pollen or anther does not dehiscent in plant”.
Line 53-55: When CMS-related ORFs are expressed and targeted to mitochondria, there is a positive and negative correlation between CMS-related ORFs and the occurrence of CMS [14]. What are these correlation? What is the importance of negative correlation between CMS-related ORFs and the occurrence of CMS?
Line 65-66: Abundant of studies reported that DNA methylation is associated with cytoplasmic male sterility. DNA methylation polymorphism in the F1 generation was higher than parents in rice. Give suitable reference for both the lines
Line 80: write heading “DNA methylation pattern of sugar beet”
Line 81-82: We collected flower buds from the sugar beet CMS line DY5CMS and its maintainer DY5O for constructing genomic DNA libraries. Is there any relationship between stage of buds with methylation level?
Line 100-101: In the three contexts, the methylation level of sterile line DY5CMS was higher than that of maintainer line DY5O. From table 2, it seems that the level of methylation was not significantly higher in maintainer line; therefore, the entire sentence need revision.
Line 125-127: DNA methylation occurs not only in CG context but also in CHG and CHH context in plants. In plants, DNA methylation occurs not only in CG context but also in CHG and CHH context in plants. Both are similar sentence; please delete anyone
Line 127-128: The genome-wide DNA methylation levels of sterile line DY5CMS and maintainer line DY5O. What is the meaning of this sentence?
Line 128-130: It was found that the genome-wide DNA methylation levels of beetroot sterile line and maintainer line were similar, and the difference in methylation levels was not significant. But at Line 100-101; it is different, please change according to the results.
Line 143-144: Different types of C base (mCG, mCHG, and mCHH) DNA methylation levels were different among different species, and even under different cell types of the same species. What is the meaning of this sentence in the results of the present study? DNA methylation levels were different among different species? Which species you are talking here? Or it is the part of discussion?
Line 175-177: DNA methylation levels above 70% are called hypermethylation levels, while DNA methylation levels below 70% are called hypomethylation levels. Please give suitable reference.
Line 201-203: DMR analysis can help us find the differentially methylated genes more quickly, which is the main purpose of the experiment. After finding DMRs, strict screening should be carried out. Is there any meaning of this sentence here?
Line 214-215: DMR can be divided into two categories: DMR-related genes and DMR-related promoters? Write this sentence before previous sentence.
Discussion part may be improved accordingly; and M&M part is fine
Comment: Manuscript need major revision

Reviewer 2 Report
Comments and Suggestions for Authors
Major Comments:
1. Therefore, this paper aims to identify the differential methylated genes in sugar beet sterile lines and maintainers by genome-wide methylation sequencing technology, conduct functional analysis, and explore the molecular mechanism of cytoplasmic male sterility in sugar beet. What do the sterile lines mean?
2. In plants, DNA methylation occurs not only in CG context but also in CHG and CHH context in plants. Do you think this is a statement?
3. The results and conclusion section explaining the findings faces the same issues observed earlier. The narrative in the results section (and conclusion) is difficult to follow, and the conclusions drawn seem significantly distant from the empirical results.
4. The discussion should focus on organized arguments, avoiding mere descriptive details without substantial meaning. A meaningful discussion should also establish connections between the study's findings and relevant theory or literature..
5. qRT PCR analysis: Need more details.
6. Clean data is obtained by filtering the offline data: need explanation.
Comments on the Quality of English LanguageEnglish is modest. Therefore, the authors need to improve their writing style. In addition, the whole manuscript needs to be checked by native English speakers.
Round 2
Reviewer 1 Report
Comments and Suggestions for Authors
Compliances from the authors are correct.
Can be accepted for publication.
.
Author Response
Dear Reviewer,
I hope this message finds you well. I would like to express my sincere gratitude for your time and effort in reviewing my manuscript titled "[Title]." Your constructive feedback and insightful comments have been invaluable in enhancing the quality of the paper.
I have carefully addressed all your suggestions and revisions, and I am grateful for your approval of the manuscript for publication. Your support and guidance have played a significant role in shaping the final version of the article.
Once again, thank you for your dedication to the peer review process.
Best regards,
Hui Wang
Reviewer 2 Report
Comments and Suggestions for Authors
The authors addressed some of my comments. But still, my major concern is: The results and conclusion section explaining the findings faces the same issues observed earlier. The narrative in the results section (and conclusion) is difficult to follow, and the conclusions drawn seem significantly distant from the empirical results.
Comments on the Quality of English LanguageNeed some minor improvements.
